# SNPs in lncRNA KCNQ1OT1 Modulate Its Expression and Confer Susceptibility to Salt Sensitivity of Blood Pressure in a Chinese Han Population

**DOI:** 10.3390/nu14193990

**Published:** 2022-09-26

**Authors:** Yunyi Xie, Han Qi, Wenjuan Peng, Bingxiao Li, Fuyuan Wen, Fengxu Zhang, Ling Zhang

**Affiliations:** Beijing Municipal Key Laboratory of Clinical Epidemiology, Department of Epidemiology and Health Statistics, School of Public Health, Capital Medical University, Beijing 100054, China

**Keywords:** salt sensitivity, acute salt loading, blood pressure, lncRNA, single-nucleotide polymorphism

## Abstract

Long noncoding RNA (lncRNA) plays an important role in cardiovascular diseases, but the involvement of lncRNA in salt sensitivity of blood pressure (SSBP) is not well-known. We aimed to explore the association of sixteen single-nucleotide polymorphisms (SNPs) in five lncRNA genes (KCNQOT1, lnc-AGAP1-8:1, lnc-IGSF3-1:1, etc.) with their expression and susceptibility to SSBP. A two-stage association study was conducted among 2057 individuals. Quantified expression of the lncRNA was detected using real-time PCR. Genotyping was accomplished using the MassARRAY System. The expression quantitative tra2it loci test and the generalized linear model were utilized to explore the function of SNPs. One-sample Mendelian randomization was used to study the causal relationship between KCNQOT1 and SSBP. Significant effects were observed in KCNQ1OT1 expressions on the SSBP phenotype (*p* < 0.05). Rs10832417 and rs3782064 in KCNQ1OT1 may influence the secondary structure, miRNA binding, and expression of KCNQ1OT1. Rs10832417 and rs3782064 in KCNQ1OT1 were identified to be associated with one SSBP phenotype after multiple testing corrections and may be mediated by KCNQ1OT1. One-sample Mendelian randomization analyses showed a causal association between KCNQ1OT1 and SSBP. Our findings suggest that rs10832417 and rs3782064 might be associated with a lower risk of SSBP through influencing the KCNQ1OT1 secondary structure and miRNA binding, resulting in changes in KCNQ1OT1 expression.

## 1. Introduction

Salt sensitivity of blood pressure (SSBP) is determined as a quantitative trait in which the blood pressure (BP) for parts of the population displays variants that parallel changes in sodium loading [1]. Studies have revealed that individuals in the population have various BP responses to salt load and display various SSBP phenotypes [2]. Individuals who exhibit elevations in BP paralleled with high salt intake are viewed as salt-sensitive (SS), whereas others are viewed as salt-resistant (SR) [2]. The Genetic Epidemiology Network of Salt Sensitivity (GenSalt) study declared that the prevalence of SS, which is viewed as a qualitative trait of SSBP, is generally 30% in North Chinese adults [2,3]. SSBP is not only the intermediate phenotype for developing hypertension but also is detrimental to the development of cardiovascular diseases (CVDs) that increase mortality [3,4]. Long-term follow-up research for normotensive and hypertensive individuals showed evidence for SSBP increasing mortality [5]. Thus, the early detection of SSBP, subsequent salt-limiting interventions, and personalized medicine may benefit from reducing the burden of CVDs.

Genes can participate in SSBP pathology via different pathways such as the renin–angiotensin–aldosterone system and ion–water channel [1,6]. Single-nucleotide polymorphisms (SNPs) are the most frequent genetic variants and are associated with gene expression, function, and diseases [7]. SNPs in some coding genes or noncoding genes have been found to be associated with SSBP. Some genome-wide association studies (GWAS) and candidate gene research have been conducted for SSBP or SS [8,9]. The GenSalt study identified eight novel loci for BP responses to dietary sodium and potassium intervention and the cold pressor test in the Han Chinese population [8,10]. Citterio et al. [11] demonstrated that SNPs located in the PRKG1 gene are associated with SSBP in Caucasian individuals with mild hypertension under the acute salt-loading test in a GWAS study. Our previous study suggested that SNPs are located in protein-coding genes such as PRKG1 and SLC8A1 [10,12]. While about 2 percent of the human genome encodes proteins, most of them are detectably transcribed in certain circumstances [11,13]. Long noncoding RNAs (lncRNAs) are defined as a type of non-protein-coding transcripts of more than 200 nucleotides [12,14]. LncRNAs have been suggested to perform several functions such as transcriptional regulation [13,15]. Our previous studies have reported the transcriptome profiles of SSH and constructed a ceRNA network to help elucidate the mechanism of SSH [14,16], and reported that lncRNAs could participate in the biological pathways in SS. We further detected that lncRNA lnc-IGSF3-1:1, SCOC-AS1, and SLC8A1-AS1 could perform as circulating biomarkers of SS [15,17].

SNPs in lncRNAs may be involved in the disease by linking to modification of the lncRNA sequence or altering their gene expression levels and influencing their regulatory capacity [18]. Additionally, SNP-caused structural disturbance within the lncRNAs could disturb lncRNAs’ molecular functions and, thus, is likely to be involved in the physiological pathways of disease [13,15]. In the present research, based on our current discovered SS-related lncRNAs, we integrated epidemiological analysis and a bioinformatics prediction method to explore the associations between five lncRNAs (KCNQOT1, lnc-AGAP1-8:1, lnc-IGSF3-1:1, SCOC-AS1, and SLC8A1-AS1) and SSBP in order to ascertain the involvement of lncRNA-SNPs in SSBP susceptibility and its potential mechanism. The flow chart for this study is shown in Figure 1.

## 2. Materials and Methods

### 2.1. Study Subjects and Sample Collection

A total of 1684 unrelated individuals from the Systems Epidemiology Study on Salt Sensitivity (EpiSS) between July 2014 and July 2016 were enrolled in this two-stage association study, the detailed information of which has been published previously [17,19]. In North China, individuals recruited from Tieling were used as discovery sets, and those recruited from Beijing were analyzed as replication sets. Written informed consent was obtained from enrolled individuals before any research-specific tests. The modified Sullivan’s acute oral saline load and diuresis shrinkage test (MSAOSL-DST) was used to assess the SSBP [10,12]. DNA samples were available for 1684 participants. Fasting venous blood samples were assembled before the assessment and utilized for serum biochemical and RNA examination for 251 participants out of the 1684 participants. The study was approved by the Capital Medical University ethics committee (no. 2013SY22) and was registered in the WHO International Clinical Trials Registry Platform (No: ChiCTR-EOC-16009980).

### 2.2. Assessment of SSBP

The determination of SSBP was conducted using MSAOSL-DST, the details of which have been previously reported [16,19,20]. All the participants were asked to pause taking antihypertensive drugs for a full day before the assessment. Each fasting participant received an oral administration of 1000 mL 0.9% saline solution within half an hour and orally took 40 mg furosemide two hours after the saline loading. After a 5 min break in the sitting position, automatic sphygmomanometers (Omron HEM-7118, Kyoto, Japan) were used to test BP two times at 1 min intervals. The mean value was computed as the final BP value. The BP was tested 3 times in the following order: before the test, 2 h after the participant finished drinking the given saline solution (acute salt-loading process), and 2 h after taking oral furosemide (diuresis shrinkage process). Systolic BP (SBP) and diastolic BP (DBP) were recorded. Mean arterial pressure (MAP) was calculated using the equation MAP = (SBP + 2 × DBP)/3 [21]. Participants with a rise in MAP of at least 5 mmHg after salt loading and (or) a reduction of more than 10 mmHg after diuresis shrinkage were viewed as SS; otherwise, they were viewed as salt-resistant (SR). SSBP phenotypes are defined as several continuous variables including MAP change 1 (MAP after the acute salt-loading process minus the baseline MAP) and MAP change 2 (MAP after the diuresis shrinkage process minus MAP after the acute salt-loading process).

### 2.3. Real-Time Quantitative RT-PCR

Five lncRNAs (KCNQOT1, lnc-AGAP1-8:1, lnc-IGSF3-1:1, SCOC-AS1, and SLC8A1-AS1) were selected based on our previous study [15,17]. RNA was extracted using a PAXgene Blood RNA Kit (cat. No. 762174, QIAGEN GmbH, Hilden, Germany) following the instructions stipulated by the manufacturer. Quantitative RT-PCR (qRT-PCR) assays were conducted to detect the levels of lncRNAs by utilizing the SYBR Green qPCR Master Mix reagent kit (MedChemExpress) on an ABI 7900HT Real-Time PCR System (Applied Biosystem, Foster City, CA, USA), and glyceraldehyde 3-phosphate dehydrogenase (GAPDH) was utilized as the internal control. The relative changes in gene expression were calculated by using the 2^−^^ΔΔct^ method [18,22]. Each sample was measured in triplicate using the average value. The primer sequences for qRT-PCR have been summarized in our previous study [15,17].

### 2.4. Candidate SNP Selection and Genotyping

Candidate SNPs within the selected 5 lncRNAs genes and their ±5 kb flanking regions were searched with the following criteria: ① Minor allele frequencies (MAF) ≥ 5%, P for the Hardy–Weinberg equilibrium (HWE) ≥ 0.05, and calling rate ≥ 95%. ② Linkage disequilibrium (LD) r2 ≥ 0.8 was selected from the 1000 Genomes CHB population (http://www.internationalgenome.org/ accessed on 15 August 2022). ③ The Genotype-Tissue Expression (GTEx) database was used to find the expression quantitative trait loci (eQTLs), RegulomeDB (https://regulomedb.org/ accessed on 15 August 2022) [19,23], HaploReg (https://pubs.broadinstitute.org/ accessed on 15 August 2022) [20,24], and 3DSNP (https://omic.tech/3dsnpv2/ accessed on 15 August 2022) of the selected functional SNPs [21,25]. Genomic DNA was isolated from 200 μL of a suspension of EDTA-anticoagulated peripheral blood leukocytes utilizing the Magnetic Beads Whole Blood Genomic DNA Extraction Kit by using automatic nucleic acid extraction apparatus (BioTeke, Beijing, China). A NanoDrop 2000 spectrophotometer (Thermo Fisher Scientific, Waltham, MA, USA) was used to measure the concentration and purity of the extracted DNA. All SNPs were genotyped utilizing the high-throughput sequencing method on the Sequenom Mass ARRAY Platform (Sequenom, San Diego, CA, USA).

### 2.5. Bioinformatics Analysis

As lncRNA-SNP may be involved in disease pathology by changing the expression or functioning of downstream target genes through several mechanisms, bioinformatics analyses were conducted to further explore their functions. The minimum free energy (MFE) structure algorithm was used to predict the RNA secondary structure [26]. The MFE estimation is used to determine the predicted structure with the lowest free energy because it is presumed that the lower the value, the more reliable and possible the structure. MFE structure predictions were calculated using the Vienna RNA package RNAfold [23,27]. The binding sites of lncRNA and miRNA influenced by SNPs were predicted using lncRNASNP2 (http://bioinfo.life.hust.edu.cn/lncRNASNP#!/ accessed on 15 August 2022) [24,28].

### 2.6. Statistical Methods

The SSBP phenotypes were defined continuously as MAP change 1 during the acute salt-loading process and as MAP change 2 during the diuresis shrinkage process in the MSAOSL-DST. We characterized the distributions of continuous variables according to the mean and standard deviation (SD) of the normal distribution variables or the median and interquartile range (IQR) for skewed distribution variables. For continuous phenotypes with a normal distribution, Student’s *t*-test was conducted to measure the differences between the two groups. The Wilcoxon rank–sum nonparametric test was utilized to test the continuous variables with non-normal distribution and rank variables. Chi-square (χ^2^) analysis was utilized to analyze Hardy–Weinberg equilibria (HWE). The effect of each SNP was calculated using additive models and allelic models. Generalized linear models were conducted to measure the associations of SNPs with lncRNAs and SSBP phenotypes which were estimated by β and 95% CI. A cumulative genetic risk score (c-GRS) was utilized to test the integrated effect of multiple lncRNA-SNPs on SSBP. The c-GRS was divided into quartiles. Multiple linear regression was conducted to analyze the association between c-GRS quartile groups and SSBP adjusted for potential confounders. To resolve multiple comparisons and control the false positives and false negatives, the false discovery rate (FDR) was used [25,29]. For lncRNA data, log2-transformation was performed. The lncRNA data used in this paper were all log2-transformed. The statistical analysis was carried out in R software (version 3.4.4).

In mediation analysis, we considered each SNP which can influence lncRNA expression level to be the independent variable, SSBP phenotypes to be the outcome, and the corresponding lncRNA to be the mediator that may explain a portion of the SSBP risk. We utilized a two-step method using the R package “mediate” [26,30]. The model-based causal mediation test was measured in two steps. In step one, a mediator model and an outcome model were fitted. The mediator model was a linear regression of log2 (KCNQ1OT1) with the SNP, age, gender, and hypertension as the predictors. The outcome model was a linear regression model for SSBP phenotypes with the following covariates: SNP, log2 (KCNQ1OT1), SNP ∗ log2 (KCNQ1OT1) interaction term, age, gender, and hypertension. After the two models were fitted, the average causal mediation effect and average direct effect were computed through a general algorithm [30]. We used 1000 iterations and *p* ≤ 0.05 was viewed as nominally significant.

With one-sample Mendelian randomization (MR) analyses, the causal association from genetically determined KCNQ1OT1 (relative expression unit) to SSBP (MAP change 1 or/and MAP change 2) was estimated by utilizing the instrumental variable analysis with two-stage least-squares regression (2SLS). The statistical analysis was conducted using the R package “AER” [31]. Quanto version 1.2.4 was used to calculate statistical power. Among our second-stage participants, the minimal MAF of the KCNQ1OT1 SNPs was 0.05. By calculating, we found that with our sample size, the power to find an OR of 1.5 was greater than 0.8.

## 3. Results

### 3.1. General Characteristics of the Enrolled Participants

This study included 1684 participants selected from the EpiSS study, and detailed information has been published in the previous paper [19]. Appendix A shows the baseline characteristics for all individuals. In all subjects, the participants who were male, drinking alcohol, smokers, and those with hypertension had a positive correlation with SSBP risk (*p* < 0.05) and a higher LDL-C level, and had a negative correlation with SSBP risk (*p* < 0.05).

### 3.2. LncRNA KCNQ1OT1, lnc-AGAP1-8:1, and lnc-IGSF3-1:1 Levels Were Associated with SSBP

Our previous paper suggested the potential role of lnc-IGSF3-1:1, SCOC-AS1, and SLC8A1-AS1 as susceptible biomarkers for SS [15,17]. We further explored the effects of these lncRNAs on SSBP. In the first stage, we detected the relative expression of KCNQOT1, lnc-AGAP1-8:1, lnc-IGSF3-1:1, SCOC-AS1, and SLC8A1-AS1 by conducting qRT-PCR among 251 individuals, and analyzed their relationship with SSBP (MAP change 1 and MAP change 2). Our study discovered that three lncRNAs (KCNQ1OT1, lnc-AGAP1-8:1, and lnc-IGSF3-1:1) were significantly associated with SSBP (*p* < 0.05): the KCNQ1OT1 expression level was positively associated with MAP change 1 and was negatively associated with MAP change 2; the lnc-AGAP1-8:1 and the lnc-IGSF3-1:1 expression level was positively associated with MAP change 1. The results are shown in Figure 2.

### 3.3. Genotype Distributions of lncRNA SNPs and Their Association with lncRNA Expression

Several studies have demonstrated that SNPs in lncRNA may affect the expression of lncRNA [28,32]. We assumed that the expression of SSBP-related lncRNAs was affected by genotypes of corresponding SNPs. A total of 13 candidate SNPs in three SSBP-related lncRNA (KCNQ1OT1, lnc-AGAP1-8:1, and lnc-IGSF3-1:1) genes were selected in this study. All SNPs were compatible with HWE (*p* > 0.05) and the MAFs of these SNPs ranged from 5 to 38%. Detailed information on these SNPs is shown in online Appendix A. Multilinear regression analysis showed that participants with the nine SNP minor alleles (rs10832417-T, rs3782064-A, rs7925578-G, rs11023840-T, rs71034996-T, rs58956504-C, rs11023582-A, rs2411884-C, and rs12577654-T) had a lower expression of KCNQ1OT1 before and after adjusting for age and gender, respectively (*p* < 0.05). There was no significant difference between lnc-AGAP1-8:1 and lnc-IGSF3-1:1 expression with different corresponding SNP alleles (rs71402704 g and rs995060-G). The results are shown in Figure 3 and Appendix A.

### 3.4. Functional Prediction for lncRNA KCNQ1OT1 SNPs

Previous research declared that lncRNA SNPs could affect the binding site efficiency for specific miRNAs and, therefore, impact the expression levels of lncRNA [29,30,33,34]. Thus, we assumed that the above nine positive SNPs could influence KCNQ1OT1 expression by changing the lncRNA secondary structure or microRNA-binding sites. MFE change, as well as local change in the structure located around the altered nucleotide, was shown in rs10832417 and rs33782064 (Figure 4). Bioinformatics analysis reported that rs10832417-T could decrease the binding efficiency of the hsa-miR-8068; rs33782064-A could increase the binding efficiency of the hsa-miR-6834, etc., and could decrease the binding efficiency of the hsa-miR-423-5p, etc. (Table 1).

### 3.5. Association Study for lncRNA KCNQ1OT1 SNPs and Risk of SSBP

The first-stage study showed that rs58956504-C significantly increased MAP change 2 (β = 2.766, *p* = 0.012) playing a protective role against SSBP using the multilinear regression model (Appendix A). However, there were no statistically significant results between other SNPs and SSBP (MAP change 1 and MAP change 2) (*p* > 0.05). In the second stage, we conducted the multilinear regression analysis on 1443 individuals. Four SNPs were found to be significantly associated with MAP change 2 (rs3782064, β = 0.762, FDR < 0.05; rs7925578, β = 0.610, FDR < 0.05; rs11023840, β = 0.779, FDR < 0.05; rs12577654, β = 0.653, FDR < 0.05). Two SNPs reached borderline significance (rs10832417-T, β = 0.547, FDR < 0.1; rs7103496, β = 0.676, FDR < 0.1). There was no statistically significant result in rs58956504 and rs11023582 after FDR correction (FDR > 0.1). The individuals in the third quartile of the c-GRS decreased the MAP change 1 by 0.710 mmHg compared with those in the lowest c-GRS quartile (*p* = 0.039), and the individual in the highest quartile increased the MAP change 2 by 0.420 mmHg compared with those in the lowest c-GRS quartile (*p* = 0.009). The results are shown in Table 2.

### 3.6. Risk SNPs Mediated SSBP through lncRNA KCNQ1OT1

We further explored the mediation effect of KCNQ1OT1 expression on the association between the above six risk SNPs and SSBP in 251 participants. Mediation models were set up with the KCNQ1OT1 for SSBP as a mediator to detect the direct and indirect effects of the SNPs on SSBP, and the results are shown in Table 3. We found significant mediating effects in KCNQ1OT1 as a mediator from four SNPs (rs10832417, rs3782064, rs7103496, and rs12577654) to both MAP change 1 and MAP change 2, respectively (indirect effect *p* < 0.05; total effect *p* > 0.05). There were no significant mediating effects for rs7925578 and rs11023840 (indirect effect *p* > 0.05; total effect *p* > 0.05). When the indirect effect is significant but the total effect is not (and no suppression is present), one likely lacks the power to identify the total effect [31,35]. Increasing the sample size will clear the issue [32,36]. As expected, four SNPs (rs10832417, rs3782064, rs7103496, and rs12577654) were found to be significantly associated with SSBP in the second stage; the results are described above.

### 3.7. One-Sample Mendelian Randomization: Observational Versus Genetic Analyses

In observational analyses, a one-unit KCNQ1OT1 increase was associated with 4.147 mmHg in MAP change 1 (*p* < 0.001) and was associated with −2.829 mmHg in MAP change 2 (*p* = 0.023). Corresponding estimates in one-sample Mendelian randomization analyses were 5.581 mmHg in MAP change 1 (*p* = 0.020) and −3.464 mmHg in MAP change 2 (*p* = 0.014), respectively, by using seven SNPs (rs10832417, rs3782064, rs7103496, rs58956504, rs11023582, rs2411884, and rs12577654) as instruments. The results are shown in Figure 5.

## 4. Discussion

Our study showed that three lncRNAs (KCNQ1OT1, lnc-AGAP1-8:1, and lnc-IGSF3-1:1) were significantly associated with SSBP and that participants with the nine SNP minor alleles (rs10832417-T, rs3782064-A, rs7925578-G, rs11023840-T, rs71034996-T, rs58956504-C, rs11023582-A, rs2411884-C, and rs12577654-T) had a lower expression of KCNQ1OT1. Four SNPs (rs10832417, rs3782064, rs7103496, and rs12577654) affected SSBP by modulating the KCNQ1OT1 expression. Among them, SNPs rs10832417 and rs3782064 in the KCNQ1OT1 gene might be associated with a low risk of SSBP through changing the KCNQ1OT1 secondary structure and miRNA binding, resulting in changes in KCNQ1OT1 expression.

SSBP, as the intermediate phenotype for developing hypertension, plays a critical role in the occurrence of CVDs. The previous studies had studied SSBP from different levels, including genomics [2,3], metabolomics [33,37], and transcriptomics [15,17], to enlighten the pathogenic mechanism of SSBP. LncRNA deregulation plays an essential role in complex diseases such as CVDs [34,38]. Nearly 90% of the phenotype-associated SNPs discovered by GWAS are located beyond the protein-coding regions and map to the noncoding regions such as lncRNA. In this study, we demonstrated that three lncRNAs (KCNQ1OT1, lnc-AGAP1-8:1, and lnc-IGSF3-1:1) were significantly associated with SSBP. The results were consistent with our previous study which illuminated that lncRNAs, such as KCNQ1OT1, lnc-IGSF3-1:1, lnc-GNG10-3:1, SCOC-AS1, and SLC8A1-AS1, had an up-regulated expression in SS compared with SR [15,17].

LncRNA-SNP may change the expression or functioning of downstream target genes through several mechanisms. Zhang et al. showed that rs7130280 in the lncRNA NONHSAT159216.1 was associated with a low risk of Behcet’s disease and uveitis and influenced the interaction between lncRNA and its target genes [30,34]. Feng et al. suggested that rs140618127 in the lncRNA LOC146880 decreased ENO1 phosphorylation by increasing the binding efficiency for miR-539-5p [35,39]. Chaoqin Shen et al. revealed that rs1317082 at lncRNA CCSlnc362 decreased the susceptibility to CRC by creating a binding site for miR-4658 [36,40]. As lncRNAs do not code for a protein, their structure is considered to be important for their function. Some RNAs can yield strong structural variation upon SNP change [37,41]. So, we performed an eQTL analysis to figure out whether lncRNA-SNPs affect the expression of the genes in which they are located and LncRNA-SNP functioning mechanisms. The binding sites of lncRNA and miRNA influenced by SNPs were predicted using lncRNASNP2. RNA secondary structure prediction methods are established in thermodynamics and, usually, the MFE structure is determined. In our study, MFE structure predictions were performed using RNAfold [33,37]. We found that participants with the nine SNP minor alleles (rs10832417-T, rs7925578-G, rs3782064-A, rs11023840-T, rs71034996-T, rs58956504-C, rs11023582-A, rs2411884-C, and rs12577654-T) had a lower expression of KCNQ1OT1. Among them, rs10832417 and rs3782064 in KCNQ1OT1 could influence the secondary structure, miRNA binding, and relative expression of KCNQ1OT1 through association study and bioinformatic methods.

Later, we hypothesized that SNP was related to SSBP through affecting KCNQ1OT1 expression. In the two-stage association study, we found that four SNPs were found to be significantly associated with MAP change 2 (rs7925578, rs3782064, rs11023840, and rs12577654). Two SNPs reached borderline significance (rs10832417 and rs7103496). Mediation analyses were conducted in the first stage. We found significant mediating effects in KCNQ1OT1 as a mediator from four SNPs (rs10932417, rs3782064, rs7103496, and rs12577654) to both MAP change 1 and MAP change 2. However, no significant associations were found in the total effects, which seems odd to explain, because either they had a suppression effect (indirect effect) which was not in our research, or they had other explanations. Kenny et al. demonstrated that the test of the indirect effect is more powerful than the test of the total effect [31,35]. As such, when total effects are not large effects, it is more likely to find indirect effects as significant than finding total effects. We conducted multilinear regression models on 1443 individuals in the second stage to detect the association (total effect) between the above SNPs and SSBP. The results have been discussed above. Generally, we found that 4 SNPs (rs10832417, rs3782064, rs7103496, and rs12577654) may affect SSBP by modulating the KCNQ1OT1 expression. It is worth noting that SNPs located in KCNQ1OT1 were related to MAP change 2, but not with MAP change 1, which indicated that SNP-induced changes in KCNQ1OT1 expression only affect SSBP during the diuresis shrinkage process, not during the acute salt-loading process.

The GenSalt study identified rs10832417 in lncRNA KCNQ1OT1 had a protective effect on mean arterial pressure response to a high-sodium diet using the GWAS method [38]. We found that the rs10832417-T/G variant in an exon of KCNQ1OT1 was significantly associated with MAP change during the diuresis shrinkage process in the second stage, which was consistent with the GenSalt result [38,42]. We proceeded to explore the potential function of GWAS’s significant SNP rs10832417 through the eQTL analysis and bioinformatic prediction, which were previously described. Our results showed that rs10832417 and rs3782064 might be associated with a low risk of SSBP through influencing the KCNQ1OT1 secondary structure and miRNA binding, and resulting in changes in KCNQ1OT1 expression. Furthermore, one-sample Mendelian randomization analysis showed that KCNQ1OT1 may have a causal relationship with SSBP using the above eQTL SNPs as instruments. KCNQ1OT1 functioned at the epigenetic level, creating a positive effect on the formation of a repressive chromatin structure, and participates in the CVD process [39,43]. KCNQ1OT1 could play an essential mediator role in endothelial cell physiologic development, and endothelial dysfunction could affect the pathogenesis of SSBP [40,44]. The regulation between KCNQ1OT1, adiponectin receptors, and the p38 MAPK/NF-kB pathway was declared [41,45], which was involved in inflammation, leading to CVDs. KCNQ1OT1 acts with miR-183-3p to up-regulate CTNNB1 in vascular smooth muscle cells (VSMCs) and subsequently influences the proliferation and apoptosis of VSMCs [42,46].

Although our results reveal that rs10832417 and rs3782064 in lncRNA KCNQ1OT1 played important roles in SSBP susceptibility through changing KCNQ1OT1 expression, there are some limitations. First, the findings were concluded by association analysis and bioinformatic prediction. Our research group plans to conduct more experiments such as a luciferase assay to validate the results of our study in the future. Next, the findings of the current research were mainly reached using human peripheral blood specimens; therefore, further investigations in animal models are required. Third, because of the number of samples in lncRNA qRT-PCR, the statistical power of the lncRNA analysis was limited. Fourth, although there is no gold-standard method to determine SSBP, the chronic dietary salt-loading protocol is relatively more accurate and stable than the MSAOSL-DST. Finally, our research was limited to Han Chinese, and our findings need to be validated in different populations. 

Generally, our present study elucidated that three lncRNAs (KCNQ1OT1, lnc-AGAP1-8:1, and lnc-IGSF3-1:1) were significantly associated with SSBP. Nine SNPs’ minor alleles (rs10832417-T, rs7925578-G, rs3782064-A, rs11023840-T, rs71034996-T, rs58956504-C, rs11023582-A, rs2411884-C, and rs12577654-T) had a lower expression of KCNQ1OT1 compared with their major alleles. SNPs rs10832417 and rs3782064 in KCNQ1OT1 were negatively associated with the susceptibility of SSBP, which might influence the KCNQ1OT1 secondary structure and miRNA binding, and result in changes in KCNQ1OT1 expression. These data highlighted a potential relationship between gene variation and lncRNAs, potentially contributing to a pathological outcome, which would be a promising pathogenic mechanism and therapeutic target for SSBP. Further functional molecular experiments of the genetic variant would be of great interest.

## 5. Conclusions

The SNPs rs10832417 and rs3782064 in KCNQ1OT1 were negatively associated with the susceptibility of SSBP, which might function through influencing the KCNQ1OT1 secondary structure and miRNA binding using bioinformatic predictions, resulting in changes in KCNQ1OT1 expression.

## Figures and Tables

**Figure 1 nutrients-14-03990-f001:**
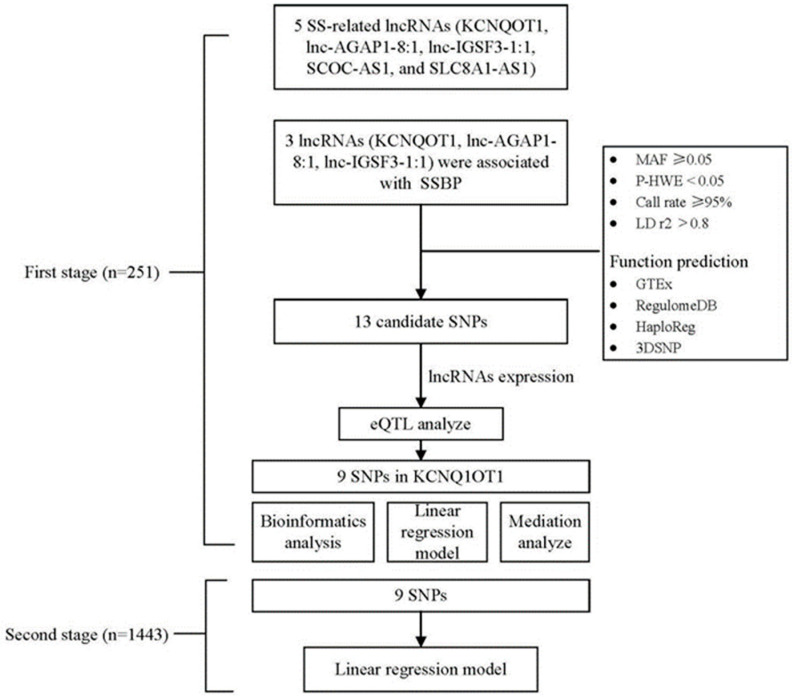
Flow chart for two-stage analysis. The workflow of the analysis includes the screening criteria and the methods. SS, salt sensitivity; SSBP, salt sensitivity of blood pressure; SNP, single-nucleotide polymorphism; MAF, minor allele frequency; HWE, Hardy–Weinberg equilibrium; LD, linkage disequilibrium; eQTL, expression quantitative trait loci.

**Figure 2 nutrients-14-03990-f002:**
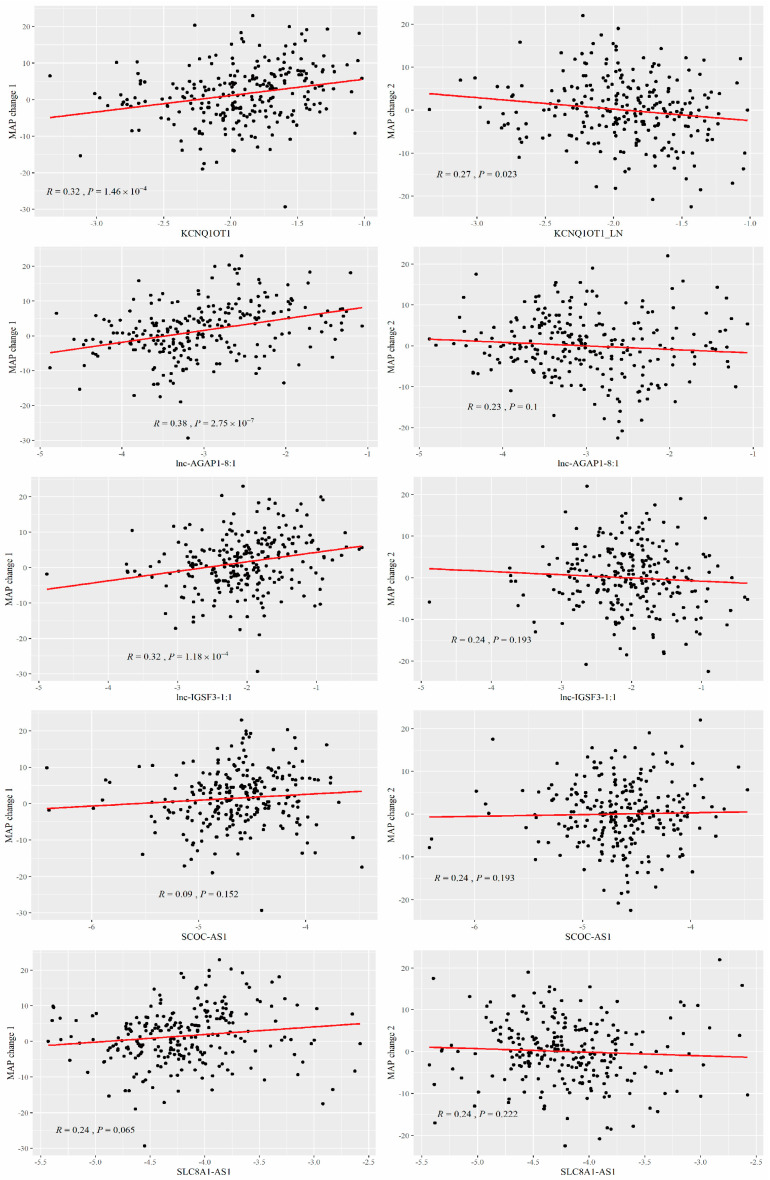
Linear regression analysis representing the association of lncRNAs with SSBP.

**Figure 3 nutrients-14-03990-f003:**
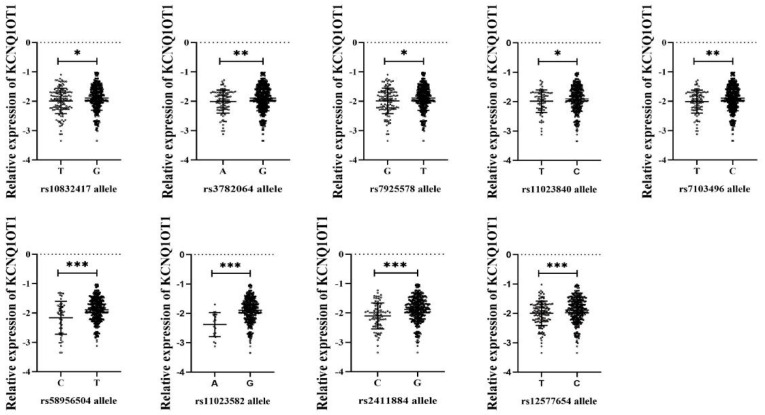
Comparison of SNP alleles regarding lncRNA KCNQ1OT1 expression level. * *p* < 0.05; ** *p* < 0.01; *** *p* < 0.001.

**Figure 4 nutrients-14-03990-f004:**
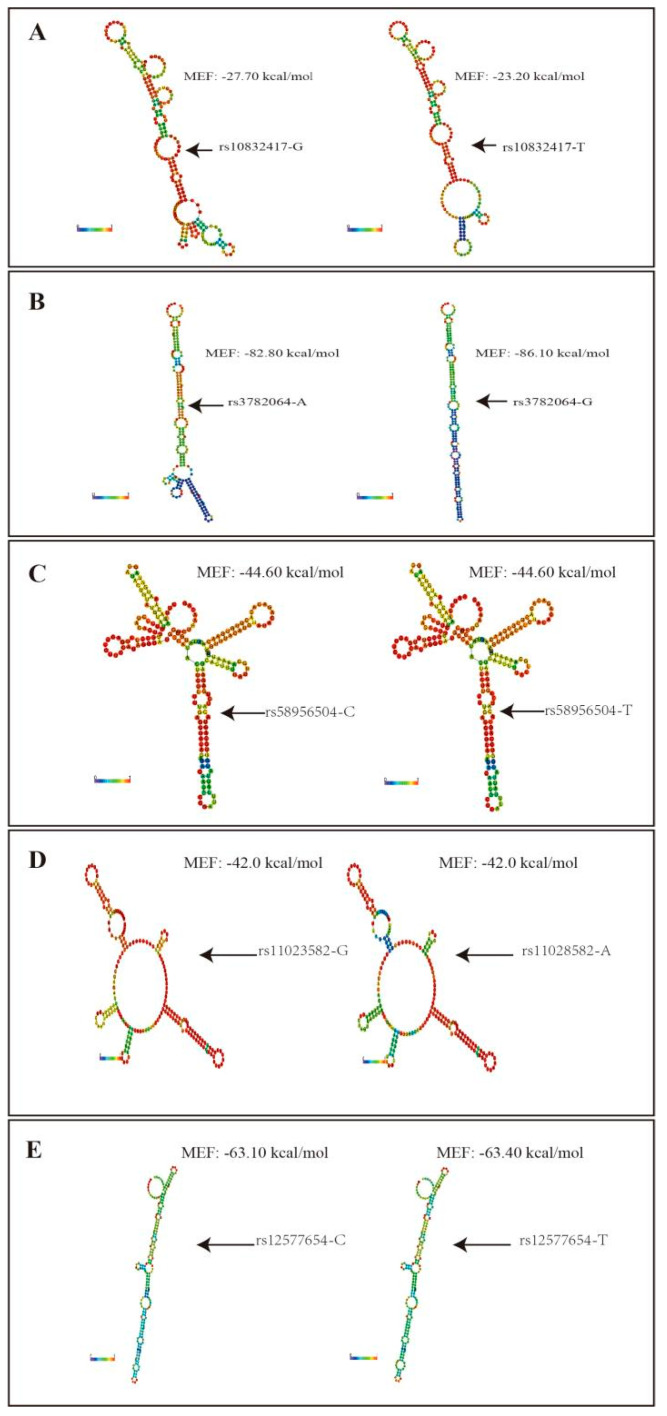
Minimum Free Energy (MFE) structure of the KCNQ1OT1. We used RNAfold. The predicted folding structures and MFE with (**A**) rs10832417 g or rs10832417-T; (**B**) rs3782064-A or rs3782064-G; (**C**) rs58956504-C or rs58956504-T; (**D**) rs11023582-A or rs11023582-G; (**E**) rs12577654-C or rs12577654-T. The structure is colored by base-pairing probabilities. For unpaired regions the color denotes the probability of being unpaired.

**Figure 5 nutrients-14-03990-f005:**
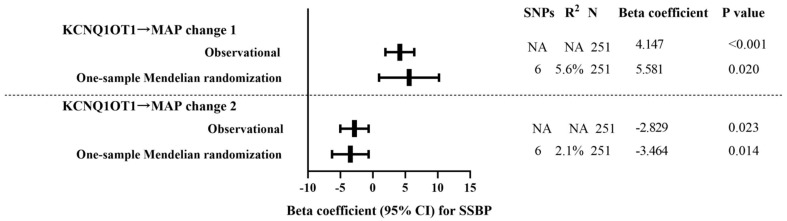
Association of observationally and genetically determined lncRNA KCNQ1OT1 and SSBP. Observational analyses used multiple linear regression multivariable adjusted for age and gender. One-sample Mendelian randomization analyses used instrumental variable analyses with two-stage least-squares regression.

**Table 1 nutrients-14-03990-t001:** The potential impact of each KCNQ1OT1-SNP on the establishment or destruction of the miRNA-binding site.

SNP	SNP Causes miRNA Target Gain	SNP Causes miRNA Target Loss
rs10832417	-	hsa-miR-8068
rs3782064	hsa-miR-6834-5p, hsa-miR-6786-3p, hsa-miR-6875-5p, hsa-miR-3126-5p	hsa-miR-3184-5p, hsa-miR-423-5p, hsa-miR-6734-5p, hsa-miR-6789-3p
rs7925578	-	-
rs11023840	-	-
rs7103496	-	-
rs58956504	-	hsa-miR-29a-5p, hsa-miR-4728-3p
rs11023582	-	hsa-miR-103a-2-5p
rs2411884	-	-
rs12577654	hsa-miR-6867-5p	hsa-miR-210-3p, hsa-miR-6790-5p

**Table 2 nutrients-14-03990-t002:** Genotype and allele frequencies of lncRNA SNPs, and genotype risks in the second stage.

	SNP	Model	Genotype	N = 1443	MAP Change 1		MAP Change 2	
					Effect Size	*p*-Value ^a^	Effect Size	*p*-Value ^a^
KCNQOT1	rs10832417	Log-Additive	TT vs. TG vs. GG	147/574/712	0.164	0.550	0.547	0.034 *
	rs3782064	Log-Additive	AA vs. AG vs. GG	45/428/925	−0.203	0.546	0.762	0.016 **
	rs7925578	Log-Additive	GG vs. GT vs. TT	146/546/701	0.157	0.572	0.610	0.019 **
	rs11023840	Log-Additive	CC vs. CT vs. TT	35/360/1021	0.259	0.468	0.779	0.020 **
	rs7103496	Log-Additive	TT vs. TC vs. CC	39/417/954	−0.294	0.409	0.676	0.036 *
	rs58956504	Log-Additive	CC vs. CT vs. TT	9/218/1206	−0.291	0.528	0.009	0.903
	rs11023582	Log-Additive	AA vs. AG vs. GG	5/141/1287	−0.363	0.532	−0.719	0.188
	rs12577654	Log-Additive	TT vs. TC vs. CC	144/640/649	−0.227	0.409	0.653	0.011 **
KCNQ1	rs2411884	Log-Additive	CC vs. CG vs. GG	59/434/875	0.063	0.063	0.137	0.708
Combined risk–effect of genotypes ^b^								
Simple-GRS	0–7 scores				Ref.	-	Ref.	-
	8–11 scores				−0.421	0.433	0.540	0.293
	12–13 scores				−0.710	0.039	0.564	0.072
	14 scores				−0.132	0.424	0.420	0.009

^a^, *p* value was calculated by adjusted age, gender, fasting blood glucose (FBG), low-density lipoprotein cholesterol (LDL-C), smoking, and drinking. ^b^, the risk genotypes used for the calculation were as follows: Simple-GRS = rs10832417 + rs7925578 + rs3782064 + rs11023840 + rs7103496 + rs12577654. * FDR-corrected *p*-value < 0.01; ** FDR-corrected *p*-value < 0.05.

**Table 3 nutrients-14-03990-t003:** ACME, ADE, and total effect and their 95% Cls between SNP and SSBP. * *p* < 0.01.

SNPs	Mediator	Outcome	ACME	ADE	Total Effect
rs10832417	KCNQ1OT1	MAP change 1	−0.392 [−0.843, −0.040] *	−0.759 [−2.257, 0.630]	−1.151 [−2.709, 0.250]
rs3782064			−0.395 [−0.873, −0.030] *	−0.882 [−2.451, 0.780]	−1.278 [−2.889, 0.400]
rs7925578			−0.317 [−0.731, 0.050]	−0.262 [−1.760, 1.230]	−0.578 [−2.112, 0.960]
rs11023840			−0.322 [−0.784, 0.050]	−1.134 [ −2.815, 0.540]	−1.456 [−3.181, 0.240]
rs7103496			−0.408 [−0.869, −0.040] *	−0.960 [−2.578, 0.660]	−1.368 [−2.976, 0.240]
rs12577654			−0.452 [−0.865, −0.120] *	−0.444 [−1.846, 1.000]	−0.896 [−2.322, 0.580]
rs10832417	KCNQ1OT1	MAP change 2	0.279 [0.017, 0.610] *	−0.255 [−1.580, 1.100]	0.024 [−1.329, 1.390]
rs3782064			0.282 [0.024, 0.600] *	−0.416 [−1.982, 1.170]	−0.134 [−1.734, 1.410]
rs7925578			0.227 [−0.041, 0.560]	−1.163 [−2.624, 0.390]	−0.936 [−2.432, 0.560]
rs11023840			0.230 [−0.040, 0.560]	−0.783 [−2.461. 0.950]	−0.553 [−2.216, 1.200]
rs7103496			0.288 [0.038, 0.660] *	−0.073 [−1.678, 1.610]	0.215 [−1.437, 1.870]
rs12577654			0.327 [0.077, 0.640] *	−0.708 [−2.089, 0.780]	−0.381 [−1.768, 1.120]

## Data Availability

The datasets generated and analyzed during the current study are available from the corresponding author upon reasonable request.

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
