# Peer review of "SNPs in lncRNA KCNQ1OT1 Modulate Its Expression and Confer Susceptibility to Salt Sensitivity of Blood Pressure in a Chinese Han Population"

_nutrients, 2022, doi:10.3390/nu14193990_

Round 1
Reviewer 1 Report
The authors of this study investigated the association of sixteen single nucleotide polymorphisms (SNPs) in five long noncoding RNA (lncRNA) genes (KCNQOT1, lnc-AGAP1-8:1, lnc-IGSF3-1:1, et al.) with their expression and susceptibility to salt sensitivity of blood pressure (SSBP).
They concluded that rs10832417 and rs3782064 might be associated with a lower risk of SSBP through influencing KCNQ1OT1 secondary structure, miRNA binding, and resulting in changes in KCNQ1OT1 expression.
The authors explored a very interesting topic. There are some minor points to discuss.
1. Lines 42-43. Please, modify the citation “Lorena Citterio” (e.g. “Citterio et al”)
2. Please, expand the background in the Introduction section (e.g. doi: 10.1016/B978-0-12-801238-3.64332-5)
3. Please, check the reference for the formula of the MAP (line 95)
4. In the text or tables, the variables with skewed distribution should be reported
5. lines 177-179. Please, rephase the sentence
6. Given the small sample size and in turn low statistical power, the authors should perform a post-hoc analysis to calculate the power of the study
7. In discussion, they should better argue on impact of the results and future perspectives
Reviewer 2 Report
The methodology for this study is adequate. The Introduction is reasonably well written and the Methods provide good detail. The Statistical Analysis section is covered comprehensively. The Results at times are difficult to follow and the Discussion is also a little confusing in parts. Overall, the major issue is the writing style, and an improvement in structuring sentences is needed. Please see my comments below.
Line 45: SNP should be defined and then abbreviated when first introduced which was in line 43.
Line 58: The following needs to be revised because the grammar is poor “…and could likely to the physiological pathways of disease.
Lines 58-62: It is extremely difficult to follow what is being described here. Please consider breaking it up into two sentences and revising your writing to improve clarity.
Line 74: China?
Lines 75-76: “..through writing..”
Line 80: “…251 out of the 1684 participants.”
Line 84: “recruiters” – would be people involved in recruiting the participants? I think you mean ‘participants’?
Line 95: “Participants with a rise…”
Line 134: “The MFE estimation was used to determine the predicted…”
Lines 176-179: Need to improve grammar here.
Line 205: “Results are shown…”
Line 240: “Results are shown in Table 2.”
Line 268: “Results are presented in Figure 5.”
Line 347: “…and participates in the CVD process.”
Liens 357-358: “Our research group plan to conduct more experiments such as luciferase assay to validate the results of our study in the future.”
Line 364-365: “….Han Chinese, and our findings need to be validated in different populations.”
Lines 370-373: This sentence is difficult to understand. Please revise.
Lines 374-378: Please revise because I do not understand the conclusions. Very poor English grammar is used here.
Round 2
Reviewer 2 Report
The corrections made have improved the quality of your manuscript.